# Using Singular Value Decomposition and Chaotic Maps for Selective Encryption of Video Feeds in Smart Traffic Management

Oussama Benrhouma [1,*], Ahmad B. Alkhodre [1], Ali AlZahrani [1], Abdallah Namoun [1] and Wasim A. Bhat [1,2]

1   Faculty of Computer and Information Systems, Islamic University of Madinah, Madinah 42351, Saudi Arabia;
    aalkhodre@iu.edu.sa (A.B.A.); a.alzahrani@iu.edu.sa (A.A.); a.namoun@iu.edu.sa (A.N.);
    wab.cs@uok.edu.in (W.A.B.)
2   Department of Computer Sciences, University of Kashmir, Srinagar 190006, India
*   Correspondence: oussama.benrhoumaa@gmail.com or oussama.benrhouma@iu.edu.sa

**Abstract:** Traffic management in a smart city mainly relies on video feeds from various sources such as street cameras, car dash cams, traffic signal cameras, and so on. Ensuring the confidentiality of these video feeds during transmission is necessary. However, due to these devices' poor processing power and memory capacity, the applicability of traditional encryption algorithms is not feasible. Therefore, a selective encryption system based on singular value decomposition (SVD) and chaotic maps is presented in this study. The proposed cryptosystem can be used in smart traffic management. We apply SVD to identify the most significant parts of each frame of the video feed for encryption. Chaotic systems were deployed to achieve high diffusion and confusion properties in the resulted cipher. Our results suggest that the computational overhead is significantly less than that of the traditional approaches with no compromise on the strength of the encryption.

**Keywords:** SVD; chaotic maps; selective encryption





## 1. Introduction

Smart-traffic management systems make use of sensors, car dash cams, street and traffic light cameras, mobile networks, and many other technologies to monitor, regulate, and respond to traffic congestion and vehicular accidents [1]. It mainly relies on the video feeds received from these technologies, which are processed by servers. Any unauthorized access to these feeds during their transmission can result in a confidentiality breach of the system. Therefore, ensuring the confidentiality of such video feeds is of paramount importance in smart-traffic management. Unfortunately, due to the huge volumes of data generated by such sources [2] and their low computational power and memory capacity, applying classical encryption schemes [3] such as Advanced Encryption Standard (AES) [4] and Data Encryption Standard (DES) [5] is not feasible. In addition, certain properties in multimedia content, such as redundancies and high correlation, makes the use of such classical encryption schemes inefficient and expensive in terms of computational time and power required [6], especially in real-time applications such as smart-traffic management.

Fortunately, chaos-based cryptography presents a solution where certain properties of chaotic systems, such as the high sensitivity to initial condition (IC) and control parameter (CP) and random-alike appearance, can be used to yield strong encryption but with less computational and memory capacity required [7]. It is relatively easy to generate such chaotic system using differential or recursive equations. Over the last two decades, various chaotic-based image encryption schemes were proposed [8–24]. However, as the volume of data increases, the computational and memory capacity requirements also increase. To overcome this problem, selective encryption can be used to select and encrypt only the most significant parts of the data [25–29]. If the selection is done properly, encryption

using chaotic maps of the selected parts of an image (or a video sequence) results in strong encryption with a significantly less requirement of computational and memory capacity for encrypting huge volumes of data in real time [30]. This selective encryption can be done in a spatial domain where only the four most significant bits (MSBs) of each pixel are selected and encrypted [31], or in a frequency domain where the image is decomposed using discrete cosine transform (DCT); then, only 25% of the frequencies are selected to be encrypted [30].

In this paper, a study on the singular value decomposition (SVD) [32] is conducted, and experiments were drawn to understand the properties of the decomposition and identify the coefficients that needs to be selected for encryption; this is done in order to reduce the amount of data that needs to encrypted given the huge size of the video feeds and the limitations in the computational power. Chaotic maps [8,30] were used in the design of the cryptosystem to increase the complexity and take advantage of the random-alike appearance. The proposed scheme is implemented and evaluated, and the results suggests that the scheme dramatically reduces the computational overhead.

The rest of the paper is organized as follows: Section 2 presents the background and related work, Section 3 discusses the proposed scheme, Section 4 describes the experiments and results, and finally, Section 5 presents the conclusion.

## 2. Background and Related Work

### 2.1. MPEG Video Sequence

An MPEG video sequence has three types of frames: (a) an I-frame that represents the most important information of the sequence, (b) a P-frame that represents a predictive frame, and (c) a B-frame that is a bi-directional frame, as shown in Figure 1.

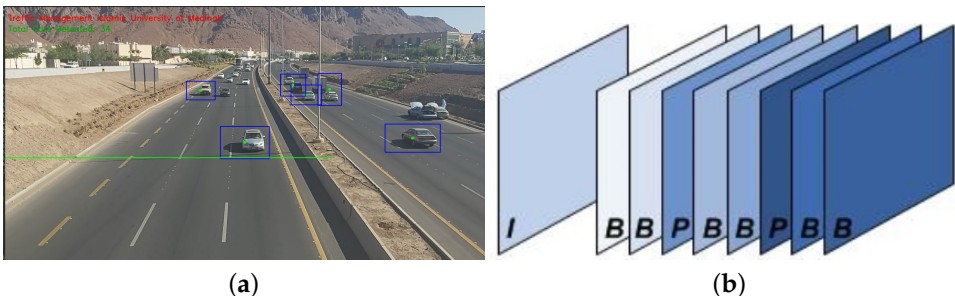

|                (**a**)                |                (**b**)                |

**Figure 1.** (**a**) An MPEG video sequence (**b**) MPEG video sequence structure.

A typical MPEG video sequence looks like the following: *IBBPBBPBBPBBIBBPBBP BB . . ..* Since the I-frame contains the most important information of the sequence, encrypting this frame results in the encryption of the whole video sequence [33]. In this paper, the extracted I-frames are treated as JPEG images, and singular value decomposition is used to extract the most relevant parts of the frame to be encrypted using PRESENT ultra-lightweight block cipher and chaotic maps.

### 2.2. Singular Value Decomposition

Given a matrix $X$ of size $m \times n$, the singular value decomposition of $X$ results in three matrices ($U$, $S$, and $V$). $U$ and $V$ are orthogonal matrices with size $m \times m$, whereas $S$ is a diagonal matrix containing values called "the singular values". Equation (1) shows the SVD results.

$$X = U * S * V^T \qquad (1)$$

To prove the importance of the diagonal matrix $S$, an image is split into blocks sized $4 \times 4$, and the svd is applied on each. As shown in Equation (2), the singular matrix for each block contains four singular values : $S_1 \ldots S_4$. Due to the huge difference between the first value and all other values, we can conclude that $S_1$ has the most significant impact on

the image. The values of the singular values for some blocks of the standard images are shown in Table 1.

$$B_i(X) = U \cdot S \cdot V^T$$

$$= U \cdot \begin{bmatrix} S_1 & 0 & 0 & 0 \\ 0 & S_2 & 0 & 0 \\ 0 & 0 & S_3 & 0 \\ 0 & 0 & 0 & S_4 \end{bmatrix} \cdot V^T \tag{2}$$

**Table 1.** Values of the singular matrix for some blocks of the standard images.

| Standard Image | Block Number | Value | | | |
|---|---|---|---|---|---|
| | | S1 | S2 | S3 | S4 |
| cameraman | 1 | 628.5056 | 16.6462 | 7.8219 | 2.7480 |
| | 33 | 474.5510 | 24.6315 | 8.2719 | 0.4573 |
| tank | 40 | 522.7868 | 14.9429 | 3.7784 | 1.1981 |
| | 200 | 527.5800 | 13.3748 | 10.0325 | 0.9129 |
| lena | 4000 | 530.3523 | 36.7818 | 8.0956 | 3.1669 |
| | 2150 | 455.2935 | 14.5609 | 6.7711 | 3.6052 |

To prove the importance of the first singular value (S1) for each block in comparison with all other singular values, multiple standard images are used (shown in Figure 2), and each image is split into blocks sized $4 \times 4$. Singular value decomposition is applied on each block, and the last three singular values ($S_2$, $S_3$, and $S_4$) are set to zeros. Then, the images are reconstructed, and the results are illustrated in Figure 3. The results prove that even with only a quarter of the singular values, we are able to reconstruct the images with acceptable quality. This compels us to conclude that if this quarter is encrypted, it will result in the encryption of the whole image, while the amount of data to be encrypted is reduced to a quarter.

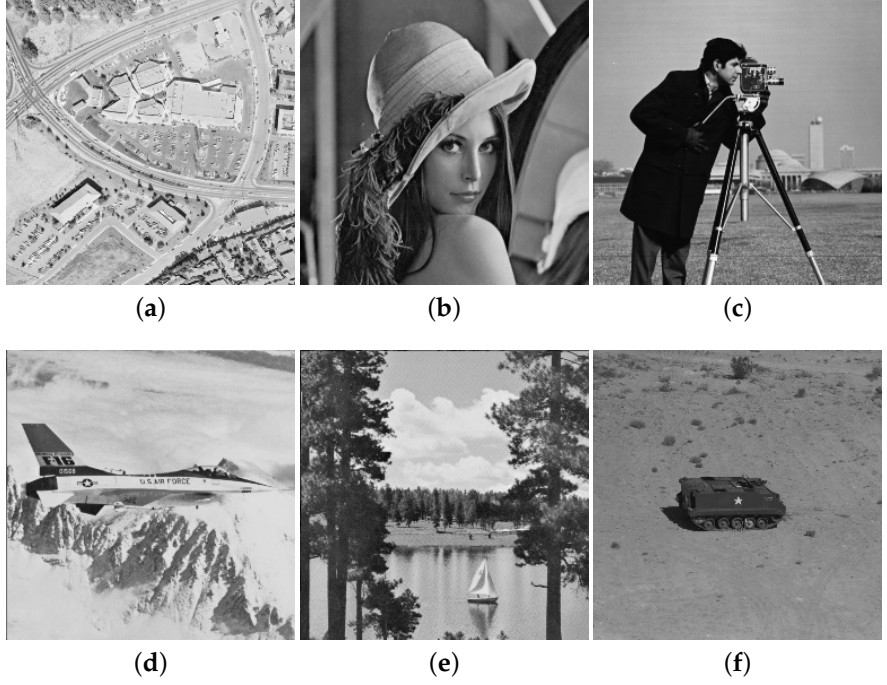

**Figure 2.** Original standard images. (**a**) aerial; (**b**) lena; (**c**) cameraman; (**d**) jet; (**e**) lake; (**f**) tank.

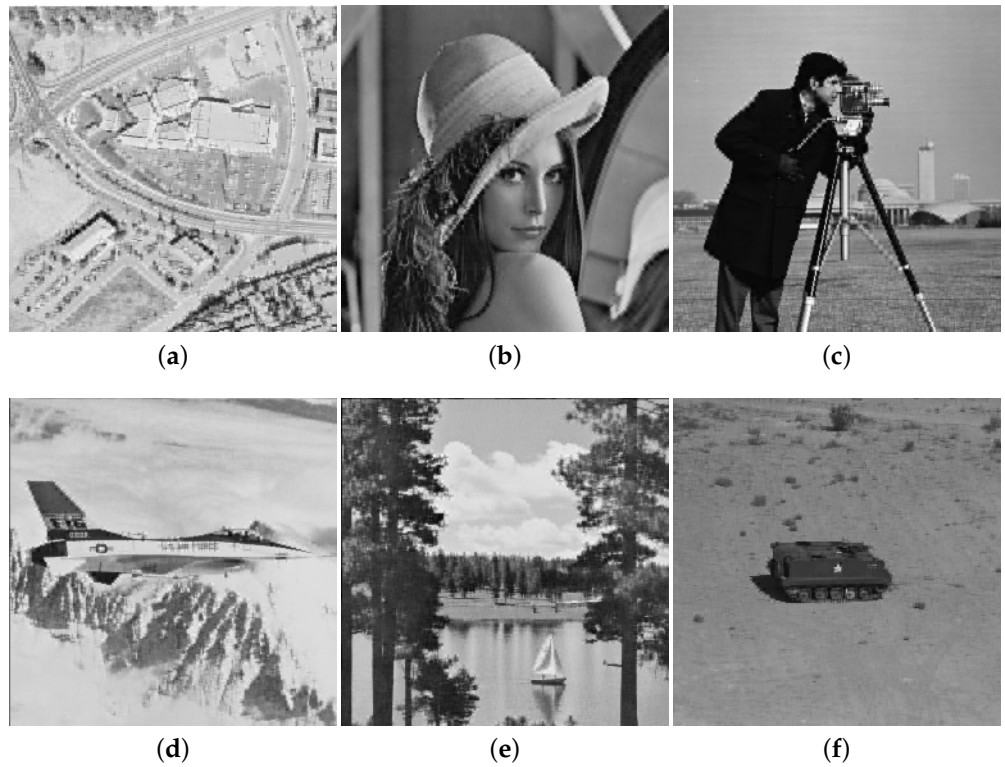

**Figure 3.** Images reconstructed with only the quarter of singular values. (**a**) aerial; (**b**) lena; (**c**) cameraman; (**d**) jet; (**e**) lake; (**f**) tank.

The structural similarity (SSIM) is a metric that measures the similarity between two images [34]. It is used in many fields such as watermarking and compression to measure the effects of distortion or compression on images. The closer the value of SSIM is to "1", the higher the similarity of the images. SSIM is defined by Equation (3).

$$SSIM(x, y) = \frac{(2\mu_x \mu_y + C_1)(2\sigma_{xy} + C_2)}{(\mu_x^2 + \mu_y^2 + C_1)(\sigma_x^2 + \sigma_y^2 + C_2)} \tag{3}$$

where:

- $\mu_x$ and $\mu_y$ represent the averages of $x$ and $y$, respectively.
- $\sigma_x^2$ and $\sigma_y^2$ represent the variances of $x$ and $y$, respectively.
- $\sigma_{xy}$ represents the covariance of $x$ and $y$.
- $C_1 = (k_1 L)^2$ and $C_2 = (k_2 L)^2$ are two variables used to stabilize a division with a weak denominator.
- $L$ is the dynamic range of the pixel values.
- $k_1 = 0.01$ and $k_2 = 0.03$.

Table 2 shows the SSIM values between the original images and the images reconstructed using only one-quarter of the singular values. The results prove that the most important information of image blocks is contained within the first singular value.

**Table 2.** SSIM values for original and reconstructed images.

| Aerial | Lena | Cameraman | Jet | Lake | Tank |
|--------|------|-----------|-----|------|------|
| 0.8007 | 0.9250 | 0.9323 | 0.9285 | 0.8900 | 0.8737 |

### 2.3. Logistic Map

The logistic map is used to take advantage of chaotic functions' pseudo-random nature. The bifurcation diagram of the function described in Equation (4) is used to determine its chaotic behavior, which is depicted in Figure 4a. This behavior can be seen when the control parameter $\mu \in [3.8, 4]$. The quasi-uniformity of the invariant natural density of the logistic map, as shown in Figure 4b, confirms this.

$$x_{n+1} = \mu x_n(1 - x_n) \tag{4}$$

Exploiting the pseudo-random behavior of the logistic map, 80 pseudo-random values are generated starting from an initial condition $x_0$ and a control parameter $\mu$; these values are then quantified to 0s and 1s based on a threshold $T$ to construct an 80-bits ling key, which will represent the secret key for the PRESENT block cipher.

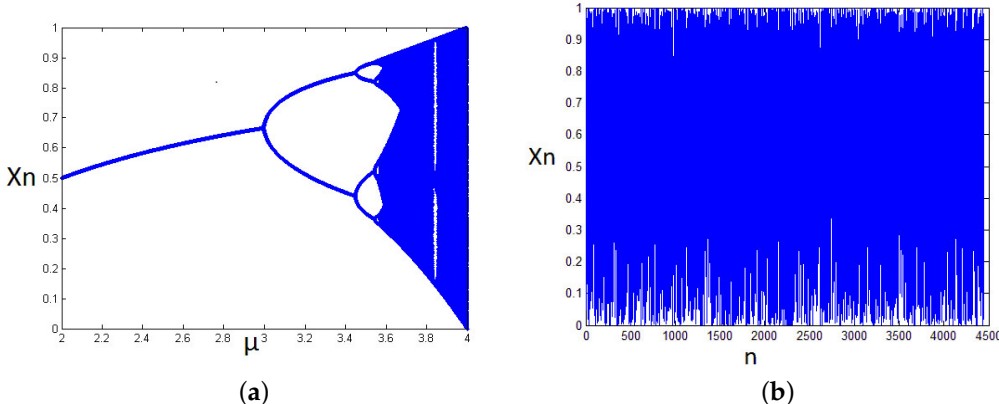

(a)          (b)

**Figure 4.** (**a**) The bifurcation diagram. (**b**) Invariant natural density.

### 2.4. The Generalized Cat Map

The generalized cat map (Equation (5)) is used to shuffle pixel positions and increase the confusion and diffusion in the resulted cryptogram.

$$\begin{bmatrix} x_{i+1} \\ y_{i+1} \end{bmatrix} = \begin{pmatrix} 1 & a \\ b & ab+1 \end{pmatrix} \begin{bmatrix} x_i \\ y_i \end{bmatrix} \bmod N \tag{5}$$

where $x_i, y_i$ represent the position of a pixel, and $x_{i+1}, y_{i+1}$ represent its new position after the iteration; $N$ is the size of the squared image.

The cat map shows a periodic phenomenon for some parameters $a$, $b$, and $N$, which means that after a certain number of iterations $P$, all the pixels return to their initial positions [35]. Figure 5 shows an example of the periodic phenomenon of a cat map when applied on a standard image. All the pixels of the image returned to their initial positions after 96 iterations.

In our scheme, the parameters used in the cat map are as follows: $a = b = 1$ and $N = 128$. The equation used is given below (Equation (6)).

$$\begin{bmatrix} i' \\ j' \end{bmatrix} = \begin{bmatrix} 1 & 1 \\ 1 & 2 \end{bmatrix} \begin{bmatrix} i \\ j \end{bmatrix} \bmod 128 \tag{6}$$

where $(i, j)$ represents the initial position of the pixel, and $(i', j')$ indicate its new position.

It is worth mentioning here that the images used in Equation (6) are of size $128 \times 128$. For larger images (say $256 \times 256$), the images are split into sub-images of size $128 \times 128$ to apply shuffling on one sub-image at a time.

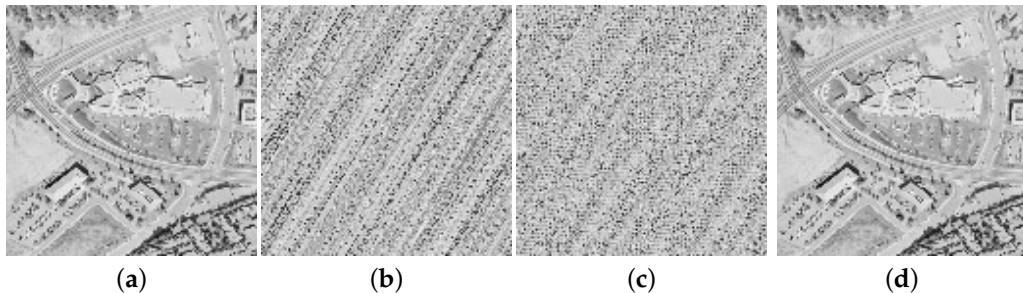

|     |     |     |     |
|:---:|:---:|:---:|:---:|
| (**a**) | (**b**) | (**c**) | (**d**) |

**Figure 5.** Periodic phenomenon of a cat map applied on a standard image. (**a**) aerial $128 \times 128$; (**b**) 3 iterations; (**c**) 50 iterations; (**d**) 96 iterations.

### 2.5. PRESENT Block Cipher

PRESENT [36] is a lightweight block cipher that consists of a substitution–permutation network [37]. The block size is 64 bits, and the key size is either 80 or 128 bits. The encryption is performed through 31 rounds. The flowchart of the working of PRESENT block cipher is shown in Figure 6.

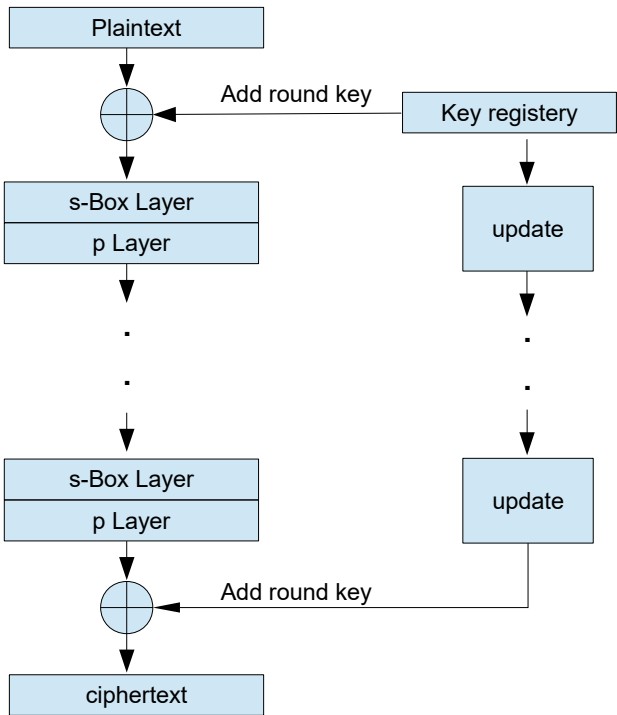

**Figure 6.** Flowchart of the working of PRESENT block cipher.

### 2.6. Selective Encryption

One of the potential techniques for ensuring image secrecy in resource-constrained devices is selective encryption. This technique selects and encrypts only significant parts of the images. If the selection is done properly, the encryption of the selected part results in the confidentiality of the whole image. The selection techniques can be performed in the spatial domain by selecting and manipulating pixel values directly or in the frequency domain by applying one of the transformations to represent the image in the frequency domain. One of the earliest contributions to selective encryption in the spatial domain demonstrated a significant reduction in computational overhead by encrypting only the most significant bits (MSBs) while leaving the least significant bits untouched [31]. Another study proposed a classification of the bit planes representing the pixels based on their significance [19]. The study encrypted only significant bits with a key stream sequence that is generated

using a chaotic pseudo-random binary number generator to evaluate it. Ref. [21] proposed a technique that initially divides a plain-text image into blocks and then calculates the correlation coefficients of each block. The random numbers generated from a skew tent map based on a predetermined threshold value were pixel-wise XORed with the block with the highest correlation coefficient values. Similarly, a stenography algorithm was proposed that consists of selecting and encrypting only 4 MSBs of the secret image to be embedded into the cover [38]. This significantly reduces the computational overhead and results in a significant visual distortion of the cover image and security of the secret image. Studies have also used discrete cosine transform (DCT2) to transform the image into the frequency domain, and then selecting only AC and 25% of DC coefficients for encryption [30]. This reduces the number of coefficients to be encrypted, thereby reducing the computational overhead. Studies have also applied discrete wavelet transform (DWT2) to divide the image into an approximation coefficient and detail coefficients, and it encrypted only the approximation frequencies [39,40]. As the approximation coefficient contains the most critical information of the image, this led to the encryption of the entire image.

## 3. Proposed Approach

The proposed approach employs SVD to select that part of each frame of video feed that contains the most significant information. This selected part of the frame is encrypted using PRESENT block cipher [36] for which an 80-bit key is generated using a logistic map. The following are the steps of the proposed approach:

Given an image M with size $128 \times 128$,

1. The image $M$ is split into non-overlapping blocks, each measuring $4 \times 4$.
2. For each block, the singular value decomposition is applied, and the diagonal matrix $S_i$ is extracted, where $i$ is the block number.

$$B_i(M) = U \cdot S \cdot V^T$$

$$= U \cdot \begin{bmatrix} S_1 & 0 & 0 & 0 \\ 0 & S_2 & 0 & 0 \\ 0 & 0 & S_3 & 0 \\ 0 & 0 & 0 & S_4 \end{bmatrix} \cdot V^T$$

3. For each block, the first singular value is extracted and stored in matrix $M_s$ with size $N/4 \times N/4$. After processing all the blocks, matrix $M_s$ is constructed, and it represents the input to encryption procedure.
4. The singular values of the matrix $M_s$ are rounded and converted to binary. Then, the binary values are transformed to a vector, which is decomposed into blocks of size 64 bits that are encrypted.
5. Starting from an initial condition $x_0$ and a control parameter $\mu$, 80 pseudo-random numbers are generated by iterating the logistic map. These numbers are quantified using a threshold $T$ to construct the encryption key $K$.
6. The encrypted blocks are reshaped, and the encrypted singular matrices are constructed.
7. The inverse singular value decomposition is applied to each block to obtain the image with encrypted singular values.
8. The pixels of the resulted images are shuffled using the Arnold cat map. The number of iterations of the cat map $t$ is considered one of the keys of the cryptosystem.

The secret keys of the scheme are: the initial condition and control parameter $(x_0, \mu)$ of the logistic map, the threshold $T$ used for the quantification step of the pseudo-random sequence, and the number of iteration $t$ of the Arnold cat map. The following are the values used in our experiments:

- $x_0$ = 987,654,326 and $\mu$ = 3.993787543;
- The threshold $T$ = 0.498754632;
- The number of iterations of the Arnold cat map $t$ = 66.

Figure 7 shows the flowchart of the encryption method, whereas Figure 8 shows the encryption results.

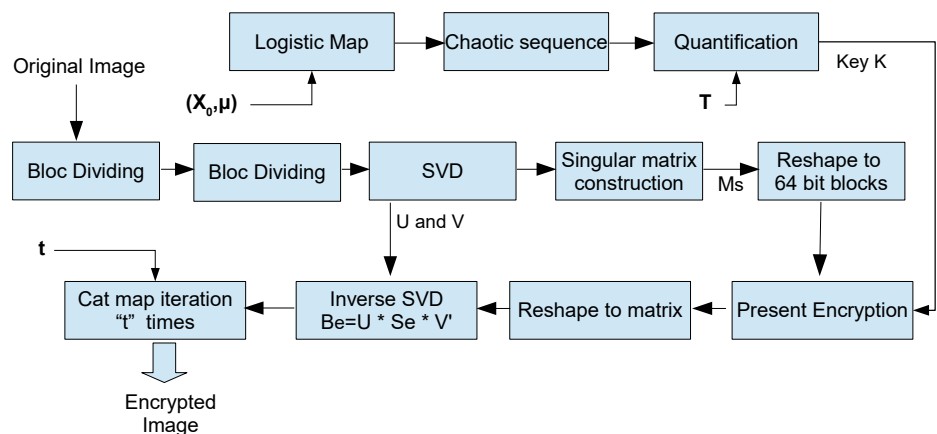

**Figure 7.** Flowchart of the encryption procedure employed.

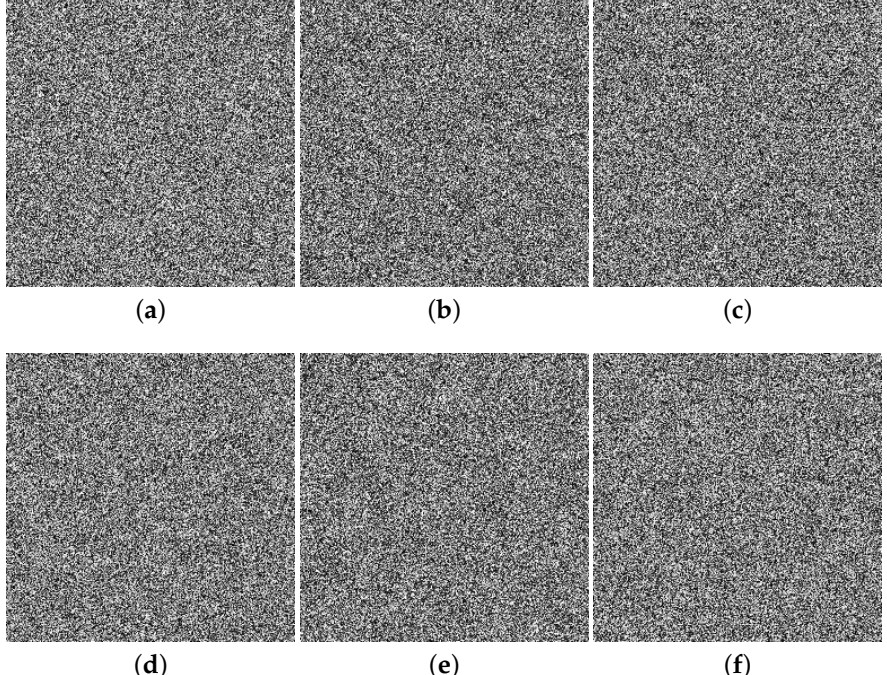

**Figure 8.** Result of encryption-procedure employed on standard images: (**a**) aerial; (**b**) lena; (**c**) cameraman; (**d**) jet; (**e**) lake; (**f**) tank.

The encrypted images can be decrypted at the receiver's end using the following procedure:

1. To restore pixel coordinates, the cat map is iterated "P-t" times, where "P" is the period of the cat map and "t" is the number of iterations in the encryption process.
2. The encrypted image $M_e$ is split into non-overlapping blocks of size $4 \times 4$.
3. Every block is decomposed using singular value decomposition, and the diagonal matrix is extracted.
4. For each block, the first singular value is extracted and stored in a matrix to construct the encrypted singular matrix.
5. The singular values of the encrypted matrix are rounded and converted to binary. The binary values are then transformed to a vector, which is is decomposed into blocks of size 64 bits that are decrypted.

6. The logistic map is used to generate 80 pseudo-random numbers, which are quantified using the threshold *T* to construct the decryption key *K*.
7. The blocks are injected along with the key *K* into the decryption procedure.
8. The decrypted values are reshaped and transformed to singular blocks.
9. Finally, the inverse singular value decomposition is applied to each block to obtain the decrypted image.

Figure 9 shows the flowchart of the decryption method, whereas Figure 10 shows the decryption results.

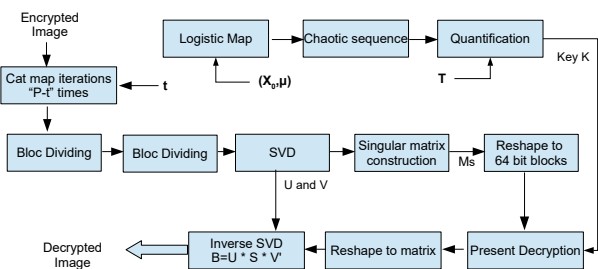

**Figure 9.** Flowchart of the decryption procedure employed.

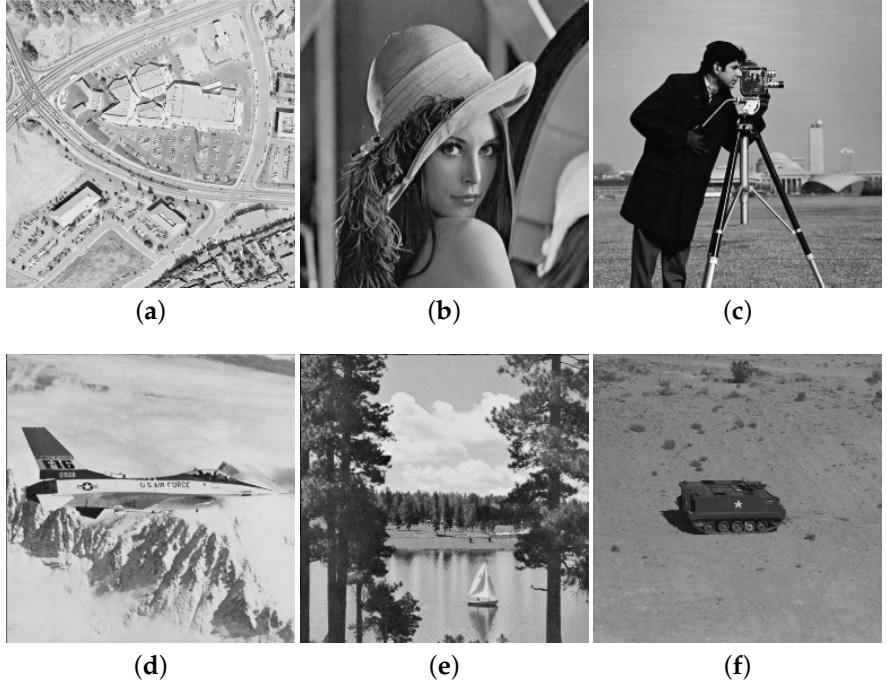

| (a) | (b) | (c) |
| (d) | (e) | (f) |

**Figure 10.** Result of decryption-procedure employed on encrypted standard images: (**a**) aerial; (**b**) lena; (**c**) cameraman; (**d**) jet; (**e**) lake; (**f**) tank.

## 4. Experiments and Results

The experimental setup used to evaluate the performance of the proposed scheme is illustrated in this section.

### 4.1. Setup

We implemented the proposed scheme using Matlab (R2016a) and executed our experiments on a Windows 10 operating system running on an Intel core i7-6500 CPU clocked at 2.50 Ghz with 8 GB of memory. Figure 2 shows the standard images used in our experiments.

### 4.2. Evaluation Metrics

To test the performance of the proposed scheme, several tests need to be conducted [41,42]. We used the following evaluation metrics:

1.  Tonal distribution of the encrypted images—It is used to see if the encrypted image is subject to any kind of statistical attack. To avoid a statistical attack, the tonal distribution of the encrypted images should be uniform.
2.  Correlation of adjacent pixels—Plain images have a high degree of correlation between neighboring pixels, which should not be present in an encrypted image. In both plain and encrypted photos, we calculate the correlation of adjacent pixels in all directions.
3.  Sensitivity to differential attacks—In this experiment, the difference between two cryptograms that resulted from the encryption of two slightly different images is measured to evaluate the cryptosystem's strength against differential attacks.
4.  Entropy—It measures the randomness in the data. An encrypted image should show a random alike appearance.
5.  Computational complexity—We calculate the time required to encrypt the image using selective encryption and full encryption. This metric evaluates the time consumed by the encryption, and it should be less for selective encryption as compared to full encryption.
6.  Key space—Key space could be defined as the theoretical set of all possible combinations. This number (key space) should be significantly large to make the brute-force attack impractical.
7.  Key sensitivity—These are experiments drawn to assess the sensitivity of the secret keys; the slightest change in the one of the keys should be diffused through the cipher.
8.  Comparison with recent encryption schemes—To validate our scheme, it needs to be compared with recent cryptosystems.

### 4.3. Results

#### 4.3.1. Tonal Distribution

An image histogram is a graphical representation of a digital image's tonal distribution [43]. Figure 11 shows histograms for plain photos, whereas Figure 12 shows histograms for encrypted images. The encrypted images' histograms demonstrate uniformity, implying that any statistical attack on the suggested system will have difficulty succeeding.

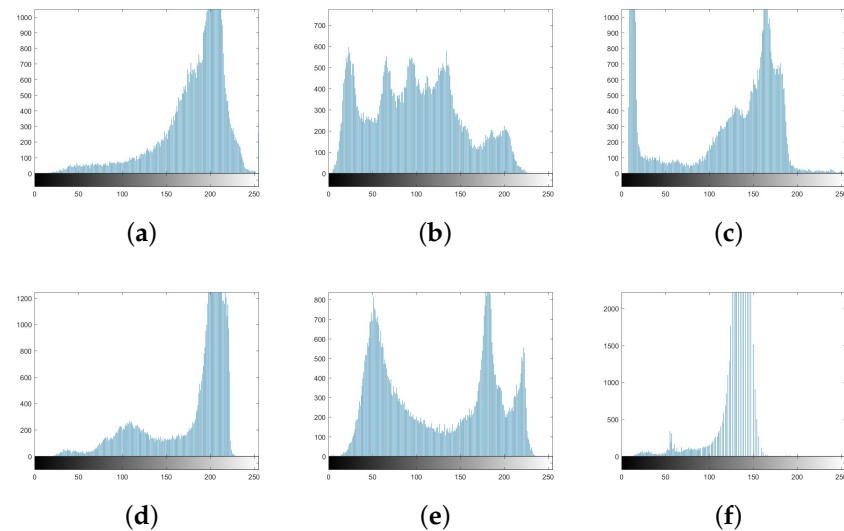

**Figure 11.** Tonal distribution: histograms of plain standard images: (**a**) aerial; (**b**) lena; (**c**) cameraman; (**d**) jet; (**e**) lake; (**f**) tank.

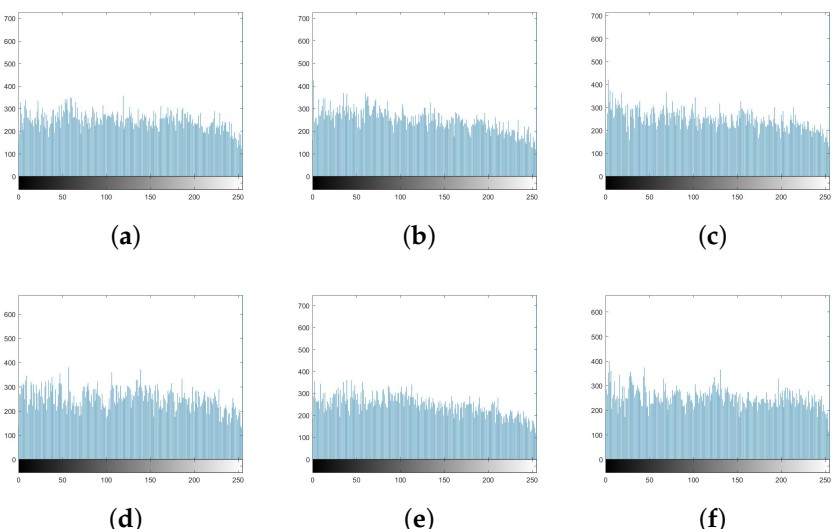

**Figure 12.** Tonal distribution: histograms of encrypted standard images that are encrypted using the proposed approach: (**a**) aerial; (**b**) lena; (**c**) cameraman; (**d**) jet; (**e**) lake; (**f**) tank.

### 4.3.2. Correlation of Adjacent Pixels

The amount of redundant information in digital photographs is extremely large, resulting in a strong correlation between adjacent pixels within the image [44]. This correlation of adjacent pixels is expected to be reduced by encryption. We used the formula in Equation (7) to compute the correlation coefficients of both plain and encrypted standard photos.

$$r = \frac{cov(p,q)}{\sqrt{D(p)}\sqrt{D(q)}} \tag{7}$$

where

$$D(p) = \frac{1}{S}\sum_{i=1}^{S}(p_i - \overline{p})^2$$

$$cov(p,q) = \frac{1}{S}\sum_{i=1}^{S}(p_i - \overline{p})(q_i - \overline{q})^2$$

- $p_i$ and $q_i$: two neighboring pixels. (horizontal or vertical).
- $S$ is the number of $(p_i, q_i)$ couples.
- $\overline{p}$ and $\overline{q}$ are the mean values of $p_i$ and $q_i$.

The correlation coefficients of both plain and encrypted standard images are shown in Table 3. The results suggest that the pixels of encrypted images show low correlation in every direction (horizontal, vertical, and diagonal), which is graphically illustrated in Figure 13.

### 4.3.3. Sensitivity to Differential Attack

Differential cryptanalysis aims to locate the "differences" in the cryptograms of two closely related plain-texts in order to find similarities that could lead to the break of the cryptosystem. To prevent that, a cryptosystem should show high plain-text sensitivity. In our cryptosystem, six plain images were used to assess the plain image sensitivity; the images were slightly altered (only some LSBs were shifted) and encrypted to compare the their corresponding ciphers. In this context, two metrics must be used: *NPCR* and *UACI*.

**Table 3.** Correlation of adjacent pixels in plain and encrypted images.

| Image | Horizontal | | Vertical | | Diagonal | |
|---|---|---|---|---|---|---|
| | Plain | Encrypted | Plain | Encrypted | Plain | Encrypted |
| Aerial | 0.6652 | −0.0018 | 0.7425 | −0.0066 | 0.6922 | 0.0084 |
| Lena | 0.9679 | −0.0062 | 0.9342 | −0.0048 | 0.8961 | 0.0077 |
| cameraman | 0.9565 | −0.0093 | 0.9333 | −0.0038 | 0.9045 | 0.0069 |
| Jet | 0.9109 | 0.0022 | 0.9023 | −0.0085 | 0.8862 | 0.0044 |
| Lake | 0.9366 | 0.0081 | 0.9340 | 0.0028 | 0.9145 | 0.0061 |
| Tank | 0.8761 | −0.0034 | 0.9195 | 0.0080 | 0.8542 | 0.0075 |

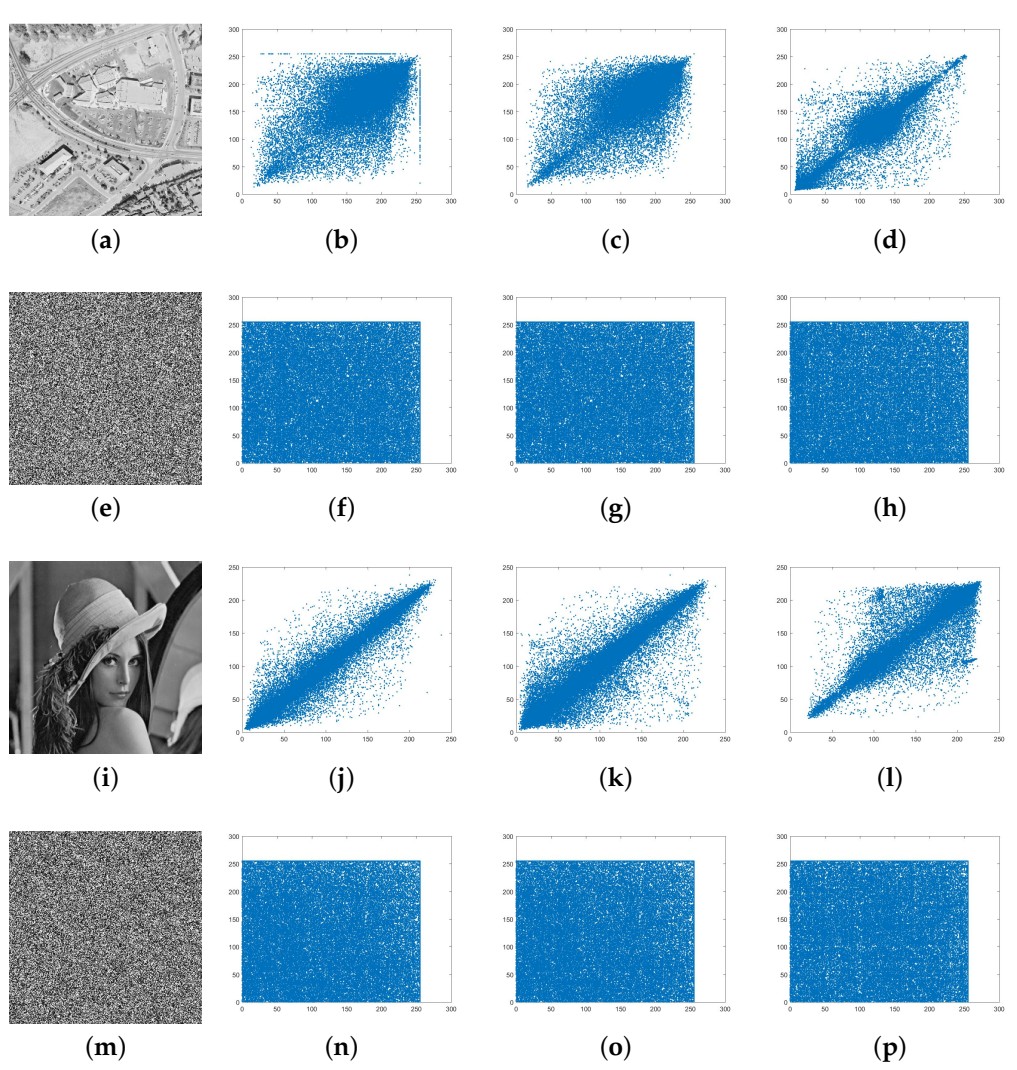

**Figure 13.** Correlation of neighboring pixels in plain and encrypted images: (**a**) aerial; (**b**) horizontal; (**c**) vertical; (**d**) diagonal; (**e**) encrypted aerial; (**f**) horizontal; (**g**) vertical; (**h**) diagonal; (**i**) lena; (**j**) horizontal; (**k**) vertical; (**l**) diagonal; (**m**) encrypted lena; (**n**) horizontal; (**o**) vertical; (**p**) diagonal.

*NPCR* and *UACI* are two metrics used to assess the similarities between two images in pixel-level comparison and in terms of average intensity change [45]. The *NPCR* is used to calculate the number of pixels that differ between the two images, which is specified by Equation (8).

$$NPCR = \frac{\sum_{i,j} R(i,j)}{M} \times 100\% \tag{8}$$

where $M$ represents the total number of pixels and $R(i,j)$ is defined by:

$$D(i,j) = \begin{cases} 0 & if \quad I(i,j) = I'(i,j) \\ 1 & if \quad I(i,j) \neq I'(i,j) \end{cases}$$

where $I$ and $I'$ are the two images in comparison and $i,j$ are the pixel coordinates. *NPCR* measured between two random images should be around 99.609375%.

The average intensity difference between the two images $I$ and $I'$ is measured using *UACI*. *UACI* is defined by Equation (9).

$$UACI = \frac{1}{M} \sum \frac{|I(i,j) - I'(i,j)|}{2^C - 1} \tag{9}$$

where $C$ is the number of bits used to encode the pixels. Between two random images, the expected value of *UACI* should be around 33.46354%.

Table 4 shows the *NPCR* and *UACI* values between two cryptograms of slightly different images. One of the experiments is shown in Figure 14. The findings show that the cryptosystem is resistant to differential attacks.

**Table 4.** Sensitivity to differential attacks—*NPCR* and *UACI*.

| Images | Aerial | Lena | Cameraman | Jet | Lake | Tank |
|--------|--------|------|-----------|-----|------|------|
| *NPCR*% | 99.6454 | 99.6515 | 99.7855 | 99.6652 | 99.7544 | 99.8711 |
| *UACI*% | 33.3288 | 32.9722 | 33.7811 | 30.1776 | 32.5110 | 30.4588 |

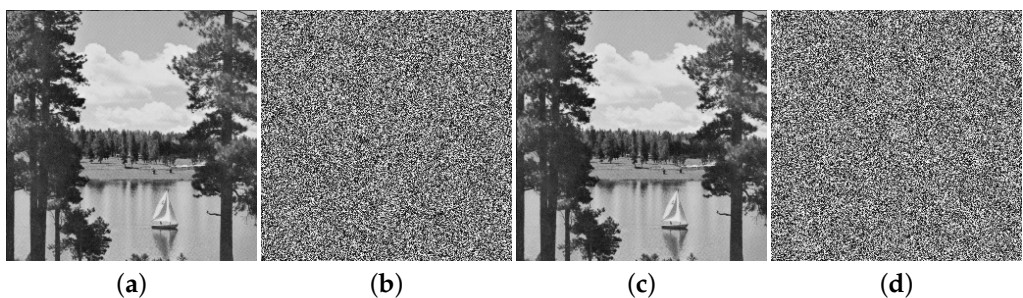

| (a) | (b) | (c) | (d) |

**Figure 14.** Plain image sensitivity: (**a**) lake; (**b**) encrypted lake; (**c**) lake SSIM = 0.9989; (**d**) encrypted lake.

### 4.3.4. Entropy

The entropy is a statistical measure of data randomness [46]. In random data, the information entropy should be around 8. The entropy of the encrypted standard images was calculated and displayed in Table 5. According to the findings, the encrypted photos contain the required entropy.

**Table 5.** Entropy results for encrypted images.

| Images | Aerial | Lena | Cameraman | Jet | Lake | Tank |
|--------|--------|------|-----------|-----|------|------|
| Enrtopy | 7.9114 | 7.9059 | 7.9178 | 7.9453 | 7.8938 | 7.9068 |

### 4.3.5. Computational Complexity

We evaluated the time consumed to encrypt an image using the proposed scheme and compare it with full encryption. In full encryption, the image is divided into non-

overlapping blocks to which SVD decomposition is applied. All singular values are then rearranged in a matrix to be encrypted. In contrast, in the proposed method, only the first SV of each block is encrypted. The results of our experiments are shown in Table 6.

**Table 6.** Computational complexity of selective encryption against full encryption.

| Image | Full Encryption | Selective Encryption |
|---|---|---|
| $256 \times 256$ tank | 31.652340 s | 3.780312 s |
| $256 \times 256$ lena | 30.875214 s | 3.010088 s |

### 4.3.6. Key Space Analysis

As mentioned in Section 3, the secret keys of our scheme are:

- The initial condition $x_0$ and the control parameter $\mu$ of the logistic map.
- The threshold $T$ used for quantification of the generated pseudo-random sequence.
- The number of iterations $t$ of the Arnold cat map

The precision of the cat map initial condition $x_0$ and control parameter $\mu$ reach $10^{-15}$. This means for these two first secret parameters, the key space reaches $10^{30}$ ($10^{-15}$ for $x_0$ and $10^{-15}$ for $\mu$). This number is almost $2^{100}$, which is a huge number that makes the key space large enough to resist exhaustive attacks. Add to that the number of iterations of the Arnold cat map, which has a size of $2^7$ and the threshold $T$, whose precision is set to $10^{-12} \approx 2^{40}$. This make the key space reach $2^{147}$. We can say with confidence that brute-force attack is not an option to attack this cryptosystem.

### 4.3.7. Key Sensitivity

To assess the sensitivity of the keys, an experiment was drawn: we are encrypting the same image using the two slightly different keys. In the experiment, we are assessing the sensitivity of the initial condition $x_0$ and the control parameter $\mu$ of the logistic map. The difference is set to $\Delta = 10^{-15}$. Figure 15 shows the results of the experiments. A structural similarity metric is used for comparison of the two resulted ciphers.

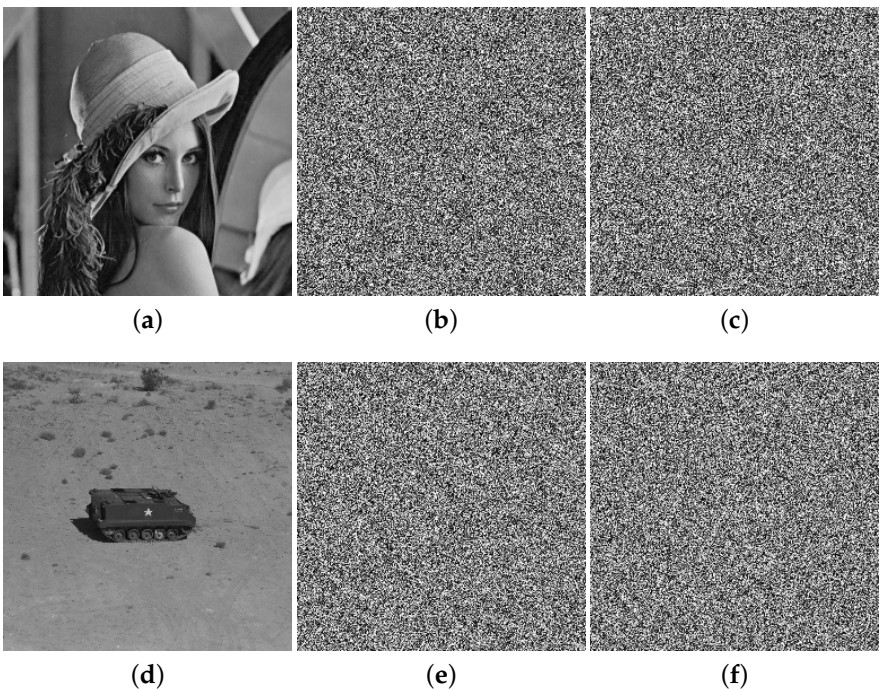

**Figure 15.** Keys sensitivity: (**a**) lena; (**b**) encrypted lena $(x_0, \mu)$; (**c**) encrypted lena $(x_0 + \Delta, \mu)$ SSIM = 0.0081; (**d**) tank; (**e**) encrypted tank $(x_0, \mu)$; (**f**) encrypted tank $(x_0, \mu + \Delta)$ SSIM = 0.0122.

As shown in the experiment, the slightest change in the key results in a huge change in the cipher.

### 4.3.8. Comparison with Related Work

Table 7 shows a comparison between our scheme and a recently proposed image encryption scheme; our schemes showed better performance in terms of NPCR and correlation values. This proves that the selection of the singular values to be encrypted is successful, and the encryption of those singular values resulted in high confusion and diffusion in the cipher.

**Table 7.** Comparison with related work.

| | NPCR% | UACI | Correlation | | | Entropy |
| | | | Horizontal | Vertical | Diagonal | |
|---|---|---|---|---|---|---|
| Proposed | 99.6172 | 30.7198 | −0.0017 | −0.0021 | 0.0068 | 7.9135 |
| Niu et al. [22] | 99.1600 | 33.0400 | 0.0478 | 0.0829 | −0.0889 | 7.9328 |
| Premkumar et al. [23] | 93.6800 | 29.6100 | −0.0586 | 0.0881 | 0.0696 | 7.4944 |
| Murali et al. [24] | 99.5210 | 33.2451 | −0.0710 | −0.0655 | −0.0953 | 7.9987 |

### 5. Conclusions

In this study, we proposed a selective encryption scheme using singular value decomposition and chaotic systems. The proposed approach aims to ensure the confidentiality of video streams originating from resource-limited devices used in a smart-traffic management system. The proposed scheme uses singular value decomposition to identify the most important parts of video frames to substantially decrease the quantity of data that needs to be encrypted. Chaotic maps were deployed, which increase the diffusion and confusion properties of the encrypted images to achieve strong encryption. Our experimental results suggest that using the proposed selective encryption results in uniform tonal distribution, low correlation between adjacent pixels, immunity to differential attack, high entropy, and low computational complexity of the encrypted images. As a result, the proposed method is appropriate for devices with minimal resources, as it provides computational efficiency due to a significantly lower amount of data processed as compared to full encryption while not compromising the strength of encryption promised by full encryption.

**Author Contributions:** Testing, O.B.; Writing original draft, O.B. and W.A.B.; methodology, A.A. and A.N.; funding and analysis, A.B.A.; Administration and revision, O.B. All authors have read and agreed to the published version of the manuscript.

**Funding:** The Deputyship for Research & Innovation, Ministry of Education in Saudi Arabia.

**Acknowledgments:** The authors extend their appreciation to the Deputyship for Research & Innovation, Ministry of Education in Saudi Arabia for funding this research work through the project number (20\14).

**Conflicts of Interest:** The authors declare no conflict of interest.

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
