# Peer review of "Using Singular Value Decomposition and Chaotic Maps for Selective Encryption of Video Feeds in Smart Traffic Management"

_applsci, doi:10.3390/app12083917_

Round 1
Reviewer 1 Report
In this paper, authors present a selective encryption scheme based on Singular Value Decomposition and chaotic maps for video applications in smart traffic management. Particularly, authors use SVD to select the most significant information of each frame of the video. Then, a recent published block cipher and chaotic maps are employed in the proposed encryption scheme. Simulations results are presented to show effectiveness and some security analysis are presented. The proposal is interesting, but there are several technical questions that must be attended.
The reviewer has the following comments:
1. Main contributions of manuscript are not clearly addressed. In Introduction Section, please specify clearly what are the mains contributions of manuscript.
2. Security analyses must be expanded. According with recent security frameworks for image encryption, please consider next references to expand and support the security analysis of proposed scheme with other analysis that can be used in propose scheme.
-
- https://doi.org/10.1142/S0218127406015970 (2006)
- https://doi.org/10.3390/e21080815 (2019)
- There is important security analysis such as secret key space, the propose scheme must define well what is the secret key and what is the key space to resist exhaustive search attack.
- Include Secret key sensitivity.
- Include Plain image sensitivity.
3. Comparison with recent schemes in literature is missed. Please include some comparisons with resents and similar schemes in literature to show advantages of proposed scheme versus literature.
4. Some figures require to improve quality. Please increase the quality of Figure 6, 7, and 9.
Author Response
Thank you for your time, all comments have been addressed.

Reviewer 2 Report
The information content and correlation coefficient in security test,NPCR and UACI need to be compared with other algorithms.
Author Response

(The authors gave the same response as above.)

Reviewer 3 Report
This paper uses SVD to identify the most important parts of video frames, reducing the amount of data that has to be encrypted significantly.
It also uses chaotic maps to boost the confusion and diffusion features of the encrypted images to achieve strong encryption.
The description of the proposed approach is very well described.
Furthermore, the experiments are well designed and can justify the conclusions.
Author Response
Thank you for your time.

Round 2
Reviewer 1 Report
In this round, the authors attended most of the comments provided by the reviewer. Nevertheless, there are some technical questions that still must be attended.
1. Technical questions in security analyses still must be attended.
a. Please see include recommended references to improve and support tje security analyses experiments:
https://doi.org/10.1142/S0218127406015970 (2006)
https://doi.org/10.3390/e21080815 (2019)
b. 2 Evaluation metrics must be attended. Particularly, 3. Sensitivity to differential attacks, its description must be reconsidered. Differential attack is related with plain image sensitivity, where two cryptograms are compared using NPCR and UACI metrics. Thus, differential attack and plain image sensitivity can be synonymous. In this sense, “the difference between the original and encrypted image” must be eliminated.
c. 3.6. Key space analysis must be attended. Please correct several typos errors in the description. The key space is not well defined. Please note that considering precision of 10^(-9) in two parameters, the possibilities are 10^(18) and not 10^(81). Thus the key space is not well defined. Use precision of 10^(-15) if it is possible for each parameter, with this, each parameter is approximate to 2^(53). According with Ref. [33] and recommended references, the key space must be more that 2^(100).
d. “4.3.7. Key space sensitivity” must be “4.3.7. Key sensitivity”.
